# Salient Features of Monomeric Alpha-Synuclein Revealed by NMR Spectroscopy

**DOI:** 10.3390/biom10030428

**Published:** 2020-03-10

**Authors:** Do-Hyoung Kim, Jongchan Lee, K. H. Mok, Jung Ho Lee, Kyou-Hoon Han

**Affiliations:** 1Core Facility Management Center, Division of KRIBB Strategic Projects, Korea Research Institute of Bioscience and Biotechnology (KRIBB), Daejeon 34141, Korea; organic2@kribb.re.kr; 2Department of Chemistry, Seoul National University, Seoul 08826, Korea; lejon0605@snu.ac.kr; 3Trinity Biomedical Sciences Institute (TBSI), School of Biochemistry & Immunology, Trinity College Dublin, The University of Dublin, Dublin 2, Ireland; mok1@tcd.ie; 4Centre for Research on Adaptive Nanostructures and Nanodevices (CRANN), Dublin 2, Ireland; 5Genome Editing Research Center, Division of Biomedical Sciences, Korea Research Institute of Bioscience and Biotechnology (KRIBB), Daejeon 34141, Korea

**Keywords:** alpha-synuclein, NMR, secondary structure propensity, pre-structured motifs (PreSMos), intrinsically disordered protein

## Abstract

Elucidating the structural details of proteins is highly valuable and important for the proper understanding of protein function. In the case of intrinsically disordered proteins (IDPs), however, obtaining the structural details is quite challenging, as the traditional structural biology tools have only limited use. Nuclear magnetic resonance (NMR) is a unique experimental tool that provides ensemble conformations of IDPs at atomic resolution, and when studying IDPs, a slightly different experimental strategy needs to be employed than the one used for globular proteins. We address this point by reviewing many NMR investigations carried out on the α-synuclein protein, the aggregation of which is strongly correlated with Parkinson’s disease.

## 1. Introduction

Alpha-synuclein (αS) is a presynaptic terminal protein that is localized at the nuclear envelope and presynaptic nerve terminals [1,2]. This small 14 kDa protein is important for the normal function and maintenance of synapses [3]. Clinically, it is strongly correlated with the pathogenesis of Parkinson’s disease (PD), a neurodegenerative movement disorder associated with the degeneration of dopaminergic neurons in substantia nigra [4], and familial early onset PD is often associated with the overexpression and mutations of αS [5,6,7]. Age-dependent motor dysfunction can be caused by neuronal fibrillar αS deposits known as Lewy bodies [8,9], the diagnostic hallmark of PD being spherical protein inclusions found in the cytoplasm of nigral neurons in the brains of PD patients.

The fibrillary aggregates of αS have a characteristic cross-β structure consisting of β-sheets, where the individual β-strands are perpendicular to the axis of the fibril [10,11,12]. These fibrillary aggregates are morphologically similar to the amyloid fibrils found in Alzheimer’s disease neuritic plaques and in deposits associated with other amyloidogenic diseases [13,14]. In addition to fibrils, advances in the structural elucidation of αS oligomers have been made recently [15,16]. Theories of (i) pore formation followed by membrane leakage [17,18], (ii) receptor-mediated mechanisms [19,20], and (iii) cellular protection by binding with extracellular chaperones [21] have been discussed in terms of the underlying molecular pathology of PD.

Although human αS is composed of 140-amino acid residues, it does not form a stable globular structure [22]. In fact, it is a well-known member of the so-called intrinsically disordered proteins (IDPs), unorthodox proteins that do not form well-defined three-dimensional structures under non-denaturing physiologcal (or near-physiological) conditions [23,24,25]. The primary structure of αS can be separated into three parts: (i) the N-terminal region (residues 1–60) has a series of 11-amino acid repeats with a conserved KTKEGV motif that, upon binding to synthetic lipid vesicles or detergent micelles in vitro, adopts a highly helical conformation [26,27,28]; (ii) the residues 61–95 that contain two additional KTKEGV repeats and the hydrophobic amyloidogenic NAC (non-amyloid-β component) region, are known to be involved in the formation of amyloid fibrils both in vitro and in vivo [14,29]; and (iii) the highly acidic C-terminal region (residues 96–140) is responsible for the overall net negative charge.

The critical early step in the fibrillation of αS is believed to involve conformational transition from the native monomeric αS into an aggregation-prone partially folded intermediate [30,31,32]. The observation that the truncation of the acidic C-terminus accelerates fibril formation in vitro [33], and the aggregated αS found in Lewy bodies has a truncated C-terminus [34], suggests that αS aggregation is slowed down by intermolecular electrostatic repulsions among the negatively charged C-terminal regions. As the partially folded oligomeric intermediates that are formed along the αS fibril formation pathway are known to be cytotoxic, we need detailed information on the conformational characteristics of the αS monomer, i.e., whether the monomer may have any peculiar conformational features that would enhance formation of oligomeric intermediates. Such knowledge should also shed light on how this protein performs its normal function.

The structural features of an IDP are described in two levels, one at a global level and the other at a local level. The global conformation of an IDP is best described as an ensemble populated with rapidly interconverting conformers [35,36,37,38]. Ensemble description is useful for understanding the overall topology of an IDP, as it provides a radius of gyration and also information on transient long-range contacts. Such ensemble representation is generally applicable to any protein and has been used not only for IDPs but for the unfolded or partially-unfolded state of globular proteins [39,40,41,42,43].

Unlike globular proteins, the ensemble structures of IDPs are not superimposable and do not converge into a single tertiary structure. Although molecular dynamics (MD) simulation alone can produce a conformational ensemble of IDPs, a more accurate ensemble is obtained when the experimental restraints from nuclear magnetic resonance (NMR) measurements, such as residual dipolar coupling constants (RDCs) [44,45] and long-range distance restraints derived from paramagnetic relaxation enhancement (PRE) experiments [40,46,47] or small-angle x-ray scattering (SAXS) experiments [48], are incorporated. An interesting point is that the ensemble conformation of an IDP (e.g., αS and tau) can be more compact than a simple random coil [46,49]. In contrast to the global conformation, the local-level conformation of an IDP is described by transient local secondary structures, termed pre-structured motifs (PreSMos) (see below) [24,25,50] that highly resemble the residual secondary structures found as a folding initiation core in the partially unfolded state of a globular protein.

## 2. Pre-structured Motifs (PreSMos) in IDPs

One prominent feature that is observed in ~70% of all IDPs or intrinsically disordered regions (IDRs) that have been thoroughly characterized by NMR is that these proteins, although intrinsically unstructured, contain transient secondary structures known as pre-structured motifs (PreSMos) [24,25]. The term PreSMos was proposed, as many different descriptions have been coined that address fundamentally the same phenomenon—that certain regions of IDPs are pre-populated with secondary structures [24,25]. PreSMos are the target-binding fragments that are primed before actual target binding. Most PreSMos are alpha-helices, but in addition there can also exist left-handed polyproline II helices (PPII), β-turns, and β-strands, and these transient structures (pre-populated only ~30% on average) become stable secondary structures in their target bound state [23,51,52,53].

Interestingly, a few IDPs such as 4EBPs (eIF4E binding proteins) and VP16 transactivation domains (TAD) have been subjected to more than one NMR investigation and the results from different investigators agree well in terms of the presence and location of PreSMos. For example, in one NMR study on 4EBP1, the residues 56–63, which form the key binding interface to eIF4E, were found to form a helix PreSMo [52]. In another NMR study on 4EBP2, a homolog of 4EBP1 with a sequence homology of ~70%, the same residues were found to form a pre-structured helix [53]. Similarly, two independent NMR groups found the same helix PreSMo in VP16 TAD that encompass the residues 442–447 and 465–483 [54,55].

Common NMR parameters obtained from the NMR studies of IDPs—chemical shifts, interproton nuclear Overhauser effects (NOEs), *R*_1_ and *R*_2_ relaxation rates (occasionally incorporated into spectral density functions), ^15^N-^1^H heteronuclear NOEs, *J* coupling constants (mostly ^3^*J*_HNHα_ associated with a backbone torsion angle *φ*), temperature coefficients of backbone amide protons, and backbone amide-water proton exchange rates—can be used to determine if an IDP possesses a PreSMo [23,24,25]. A deviation of chemical shifts from random coil values indicates the presence of a secondary structure.

Short-range interproton NOEs, such as intraresidue d_αN_(i, i) and sequential d_αN_(i, i+1)-type NOEs, are commonly observed in IDPs, whereas sequential d_NN_(i, i+1), medium-range d_αN_(i, i+2), d_αN_(i, i+3) and d_NN_(i, i+2) are observed when a PreSMo is present. The ratio of sequential d_αN_(i, i+1) to sequential d_NN_(i, i+1) NOEs [56] and that of sequential d_αN_(i, i+1) to intraresidue d_αN_(i+1, i+1) NOEs [57,58] are excellent measures of the backbone torsion angles. Thus, the combined analysis of different types of interproton NOEs can show if an IDP contains a locally ordered secondary structure. However, one should be aware that relatively weak interproton NOEs are observed in IDPs as the secondary structures in IDPs are of a transient nature. In addition, long-range interproton NOEs are absent in IDPs, as IDPs lack the stable topology that leads to such NOEs.

Whereas interproton NOEs provide short-range (< 5Å) information, paramagnetic relaxation enhancement (PRE) provides long-range (up to ~25 Å) information that may be present in IDPs [46,59,60]. Furthermore, larger spin-spin relaxation rates (shorter *T*_2_ values) are observed for the residues forming a locally-ordered segment, i.e., a PreSMo [23,25,52,61,62]. Similarly, a locally ordered protein backbone ^15^N-^1^H amide bond generates a positive ^15^N-^1^H heteronuclear NOE value. In addition, ^3^*J*_HNH__α_ coupling constants of 6 Hz or lower will be observed for helix-forming residues, while ^3^*J*_HNH__α_ values larger than 8 Hz will be observed for β-type conformations [63]. A small (< 5 ppb/deg) temperature coefficient of a backbone amide proton suggests that the proton is involved in hydrogen bonding, indicating a transient helix or β-type structure [23,54,62]. Residual dipolar couplings (RDCs) provide information on the structure and dynamics of bond orientations and are measured to assess the conformational details of IDPs [44,59].

Chemical shifts possess information on secondary structures and many analysis tools aim at assessing the conformation of IDPs from chemical shifts. First, the secondary structure propensity (SSP) algorithm uses the protein backbone chemical shifts (^13^C^α^, ^13^C^β^, ^13^C’, ^1^H^α^, ^1^H^N^, and ^15^N) to generate a residue-specific score ranging from 1 to –1, which corresponds to the fully α-helical and β-sheet structure of a well-ordered protein, respectively [64]. As the polyproline II helix is an important structural motif in IDPs, the δ2D algorithm estimates the relative population distribution of α-helices, polyproline II helices, β-sheets, and random coils from the backbone chemical shifts of IDPs [65]. If additional NMR parameters such as multiple *J* coupling values and local interproton NOEs are available for IDPs, maximum entropy Ramachandran map analysis (MERA) can be used to estimate the relative distribution of their dihedral angles on a Ramachandran map [66].

As mentioned in the introduction, another class of methods provides representative ensembles of IDPs. A statistical random coil generator (e.g., TraDES [67], Flexible-Meccano [68]) creates a pool of random conformers and an ensemble selection algorithm (e.g., ENSEMBLE [69], ASTEROIDS [70], BEGR [71]) chooses a subset of ensembles that best matches the experimental results. We used the SSP algorithm to estimate secondary structures as it is simple but powerful and only the backbone chemical shifts of αS were available for comparative analysis (Table 1).

SSP is based on the calculation of secondary chemical shifts, i.e., the difference between the measured and random coil chemical shifts (RCCS). We have used the corrected shifts for IDPs, POTENCI RCCS, for SSP analysis as POTENCI takes into account the effect of neighboring residues as well as the experimental conditions such as temperature, pH, and buffer conditions [72]. Our criterion to use SSP values is based on the lack of reporting of all measurable NMR parameters mentioned above, whereas for every IDP investigation, the backbone resonances are assigned (although only some are deposited in the biological magnetic resonance bank (BMRB) database) (Table 1). As such, we have summarized six reports in Figure 1 even though there have been more NMR studies on αS. As ^1^H^N^ and ^15^N chemical shifts are sensitive to experimental conditions and ^13^C chemical shifts can be easily re-referenced and are good indicators of α-helices and β-sheets, we only used Cα, Cβ, and C′ chemical shifts for the SSP analysis (Figure 1).

## 3. Inconsistency

An early study by Eliezer et al. [28] described that the Cα chemical shifts for residues 6–37 in αS deviated more than ~0.3 ppm from RCCS, which indicates the existence of a helix PreSMo. In addition to this transient helix, four more transient helices, centered around residue numbers 44, 60, 84, and 100, were noted. The SSP values in Figure 1 seem to agree with this regarding the most prominent N-terminal helix formed by residues 10–30, even though the degree of pre-population of this helix does not match quantitatively among one another. Eliezer et al. [28] also showed that there are four more transient helices covering the NAC region. However, it was difficult to find such additional transient helices in our SSP analysis.

In Figure 1, two cases (Figure 1b and c) show two weak helices around residues 50 and 90. The SSP data obtained at low pH (Figure 1b) peculiarly show that there is an additional helix around residue 130 at the C-terminus, in contrast to the other five results, which have a β-type transient structure at the C-terminus. However, caution is advised when interpreting these results because, for example, the N-terminal helical propensities that are observed in all SSP plots (Figure 1) are not evident in the *R*_2_ relaxation data [58,81]. Not every PreSMo shows a faster *R*_2_ relaxation rate [23,25,52,61,62]. On the other hand, the concerted measurement of ^3^*J* couplings implies that the *φ* angles have a very small helical tendency in the N-terminal 10–30 residues when the residue-specific average *φ* angles were compared in αS for each amino acid type [89].

An interesting observation is that the in-cell NMR data (Figure 1c) shows a higher population of the N-terminal helix than the *in vitro* results. This is probably due to differences in the sample conditions between in-cell NMR measurements and *in vitro* experiments, e.g., crowding effect, presence of lipid membranes. When αS was purified after it had been deliberately (by co-expressing an enzyme) N-terminal acetylated in *Escherichia coli* (*E. coli*) cells, the N-terminal residues of αS possessed enhanced helicity [78]. As it is known that αS can be N-terminal acetylated during and after translation in human cells, internally expressed or externally introduced αS in these cells is likely in an acetylated state [60]. However, because in-cell NMR data used to generate Figure 1c were obtained inside *E. coli* cells without enzymatic acetylation, we can safely rule out the effect of acetylation on αS conformation. 

Another possibility is that the higher N-terminal helicity observed in this in-cell report is simply due to the fact that only carbonyl chemical shifts were used in computing the SSP scores in the N-terminal region as Cα and Cβ chemical shifts were not available. In addition to the discrepancy in the SSP values from different NMR studies (Figure 1), there is an intriguing point regarding the presence of the N-terminal transient helix. When αS was investigated by the δ2D algorithm, no pre-structured helix around residue 25 was found at the N-terminus [78]. Therefore, we have applied different computation tools to interpret chemical shifts in terms of the αS conformation. There were small-but-significant differences when different computational tools were employed (Figure 1), even when the same ^13^C chemical shifts were used as an input.

As the Cα secondary chemical shift is by itself a good indicator of secondary structure, whose value is positive for α-helices and negative for β-type structures, we first compared the SSP scores to Cα secondary chemical shifts (Figure 1). They showed similar trends, although the SSP scores showed less fluctuations among adjacent residues. This observation can be ascribed to the algorithm of SSP, which averages the secondary chemical shifts from *i*-2 to *i*+2, and to the combined analysis of different nuclei that would reduce the observed error [64]. Next, the SSP scores were compared with the δ2D results using the same ^13^C chemical shifts as an input (Figure 1). Some secondary structures were commonly observed in the two cases, as in the α-helix at residues 3–6 and β-sheet at the C-terminal region of BMRB 6968 and BMRB 25227.

However, the secondary structure patterns do not generally match, particularly for the α-helix (residues 10–30) that is clearly observed in the SSP analysis. This can be ascribed to the small populations of secondary structure in αS, redistribution of the α-helix population in SSP into the α-helix and PPII in δ2D [65], and the different RCCS employed in the two methods, namely, POTENCI for SSP and CamCoil [90] for δ2D. Taken together, caution is advised when interpreting chemical shift data in terms of conformation. This is true even when using NMR data other than chemical shifts, although collective analysis of independent data from NMR and other experiments would aid in accurate description of IDP conformation.

In NMR studies of globular proteins, slight differences in the NMR sample conditions (protein concentration, buffer, temperature, pH, etc.) do not significantly influence the overall 3D structure. The same is true even for IDPs as was seen in VP16 TAD and 4EBP1/2; slightly different sample conditions did not influence the results in terms of the presence and/or location of PreSMos. Then why do the results of different NMR studies on αS conformation not completely agree regarding the location and the degree of pre-population of transient structures?

## 4. The Effect of Environmental Conditions on αS Conformation

We believe that several factors, as described below, should be scrutinized prior to an NMR investigation of aggregation-prone proteins, such as αS. 

### 4.1. Protein Concentration

In the early days of protein NMR experiments, often a very high (>10 mM) sample concentration was used in order to compensate for poor signal-to-noise ratios in low-field (<5 Tesla) NMR spectrometers [56]. Thus, a high concentration sample was inevitably used by NMR pulse sequence developers [91,92,93] to test their pulse schemes. The preparation of highly-concentrated protein samples was possible only because the investigators wisely chose a highly-soluble protein, e.g., bovine pancreatic trypsin inhibitor (BPTI). The measured NMR parameters with highly soluble proteins conformed well to, and complemented existing knowledge, e.g., the 3D structures known by x-ray crystallography or the properties that can be deduced from such structures. 

In subsequent protein NMR studies, sample concentrations have decreased over the few decades to such an extent that they are now comparable to, or lower than ~1 mM (Table 1). In most investigations, the importance on whether a particular protein is truly a monomeric state cannot be overemphasized. Measuring the concentration dependence of mean residue ellipticity θ in circular dichroism experiments, and showing that this relationship is linear around the NMR sample concentrations, is one way of demonstrating the monomeric nature. Alternatively, analytical ultracentrifugation can be used to detect oligomerization in protein samples. A change of the NMR line widths and diffusion rates can also indicate the presence of oligomeric species.

Ensuring the monomeric state of a protein during data acquisition becomes critical when we deal with any protein with aggregation tendencies, such as αS. All SSP scores in Figure 1 show that the C-terminus of αS has a β-type structure except for the low pH case in Figure 1b. Is it possible that protein concentrations of 300 µM–1.7 mM impose a fibrillar β-type conformation in αS? Eliezer et al. [28] used a ~100 µM protein sample and observed only the α-helical propensities for αS. Does this suggest that NMR experiments at a very low protein concentration (<50 μM) are required to assess the truly monomeric state?

### 4.2. pH

The pH of a sample is also an important factor that can influence the conformation of proteins. In the case of globular proteins, provided that the charged states of surface-exposed amino acids have no impact on the overall globular topology, the 3D structures determined at different pH values are known to be quite similar. In the case of IDPs, however, most residues are fully exposed to the solvent. The charged states of surface hydrophilic residues are very likely to influence the topology, not to mention the local structures. An exception would be the residues that form PreSMos as they can transiently form hydrogen bonds and hence experience slow backbone amide-water proton exchange. In an extreme case of the spring-loaded mechanism adopted by influenza hemagglutinin, a flexible loop undergoes a drastic conformational change to become a helix that induces viral fusion with the cellular membrane [94]. Cho et al. examined the conformations of αS at pH 3 (Figure 1b) and pH 7.4 and the two were quite different with respect to the helical content of both C- and N-terminal parts [76]. This calls for an argument that the pH of an αS sample needs to be controlled with caution.

### 4.3. Temperature

For globular proteins which have a relatively rigid backbone topology, 3D structures determined at different temperatures are not grossly different. For IDPs, like αS, with a tendency to aggregate, performing NMR experiments at 277 K (Figure 1c,e) may lower the hydrophobic effect and thereby lead to conformational ensembles that are different from those found in the cellular environment. For globular proteins, performing an NMR experiment at temperatures that are within a non-denaturing range is acceptable. Yet, for an IDP whose oligomerization is directly related to the pathology of PD, temperature can have a significant effect on its conformation.

### 4.4. Buffer and Ionic Strength

Since the thermodynamic stability of globular proteins allows them to maintain their structural homeostasis, changes in the solvent buffers are considered unlikely to pose a problem in 3D structure determination. This may not be true for IDPs, as the counterion shielding of polar residues on the surface will influence the conformations of IDPs to a greater extent than of globular proteins. In one investigation, measurements done in rather high ionic strength (200 mM) and high temperature (42 °C) conditions (Figure 1f) resulted in the degree of pre-population of the N-terminal helix to be lower than for others. Note that the effect of ionic strengths on the protonation states of side chains is compensated by using POTENCI RCCS.

### 4.5. Lipid Membranes

The functional and pathogenic role of αS is closely related to its interaction with lipid membranes. For example, αS is involved in membrane remodeling [95], clustering [96], and maintaining the pool of synaptic vesicles [97], and chaperoning SNARE-complex assembly [98]. In addition, fibril formation of αS is influenced by lipid interaction, where the promotion and inhibition of αS aggregation depends on lipid conditions [99,100]. Therefore, it is important to investigate the detailed mode of interaction between αS and lipid membranes. When doing so, we wish to note the importance of being aware of the incomplete removal of detergents and lipids during the purification of the αS protein, as they can cause small line broadening in NMR resonances. Thus, lipids should be completely removed for studies on free αS, or the lipid conditions should be precisely controlled for studies on the interaction between αS and lipid membranes.

The αS-lipid interaction has been monitored by many biophysical tools. As there is a strong correlation between the extent of αS-lipid binding and lipid-induced αS helicity, circular dichroism has been the most popular method to measure αS-lipid binding [28,101]. In addition, fluorescence correlation spectroscopy [102] and fluorescence anisotropy [27] methods have been used to measure the change of translational and rotational diffusion of αS, respectively, in the presence of lipid vesicles. However, residue-level analysis can only be performed by NMR spectroscopy.

In the initial work by Eliezer and coworkers [28], interactions between αS and lipid vesicles were monitored by NMR spectroscopy. They observed a complete disappearance of residues 1–100 in the protein backbone ^1^H-^15^N correlation spectra with a molar excess of SDS micelles or small unilamellar vesicles (SUV) over αS, which indicates that the N-terminal region of αS binds tightly to the vesicle, while the C-terminal tail is not bound to the vesicle and preserves its disordered conformation. Furthermore, the structural details of αS bound to SDS micelles were monitored by NMR spectroscopy [103]. Residual dipolar couplings were measured to determine the two (V3-V37 and K45-T92) anti-parallel curved helices connected by a linker when αS was bound to a micelle.

In the presence of a small amount of SUV composed of phospholipids (lipid molecule/αS < 5, lipid vesicle/αS < 0.001), the condition that mimics the synaptic environment, due to the high concentration of αS at the synapse, only the change in intensity and line width of αS backbone NMR resonances was observed without any perturbation in chemical shift values [27]. Although a chemical shift analysis cannot be performed to assess the bound-state conformation, the residue-specific intensity profiles showed that the residues 1–100 were bound to the vesicle with multiple tight binding modes, whereas the 40 C-terminal residues remained flexible, as in the high lipid concentration condition. In addition, transferred NOE experiments showed that the 100 N-terminal residues form a helical structure upon binding to SUV [27]. The lysine sidechains of αS are protected from acetylation by N-succinimidyl acetate in a residue-position and in a lipid-concentration dependent manner, supporting the presence of multiple binding modes between αS and lipid membranes [104].

We can also think of a mechanism on how αS strongly binds to lipid membranes. Whether or not αS contains a transient helix at its N-terminus would change its binding mode to lipid membranes, as well as the associated thermodynamic and kinetic parameters. In the early days of IDP research, when IDPs were viewed to be completely unstructured without any trace of transient secondary structures, only the induced fit mechanism involving for example, a coil-to-helix transition [105], was used to explain IDP-target binding. With knowledge that dozens of IDPs (~70% of the IDPs characterized) contain transient secondary structures termed now as PreSMos (Pre-Structured Motifs) [24,25,50], an alternative target binding mechanism of IDPs, conformational selection, is now widely considered [106,107] (Figure 2).

## 5. Summary

In this review, we have presented secondary structure propensities calculated from the chemical shifts of αS deposited in the BMRB database [108]. Although the calculated propensities appear similar (Figure 1), there are subtle-yet-significant differences among the secondary structures and the degree of pre-population of the secondary structures. For example, there is a stronger helical propensity near the N-terminus of αS under cellular conditions (Figure 1c), whereas the pH 3 condition introduces a transient helical structure at the C-terminus (Figure 1b). SSP is a common tool; however, it is one of several methods to assess secondary structures in IDPs. Development of better experimental methods and more accurate analysis tools seem to be needed to improve the accuracy for describing IDP conformations. 

Certain IDPs, including αS, are relatively more prone to aggregation than globular proteins. It is possible that αS represents one of those IDPs for which carrying out an NMR structural investigation should be performed with much more precaution than one might think, e.g., under a well-controlled condition resembling the native cellular settings as much as possible [109]. Certain sample conditions could modulate IDP conformations as exemplified here for the αS protein. Notwithstanding the plethora of previous S NMR investigations, it remains a tempting challenge to perform further investigations on αS under “authentic” conditions in order to obtain the genuine conformation of monomeric αS.

## Figures and Tables

**Figure 1 biomolecules-10-00428-f001:**
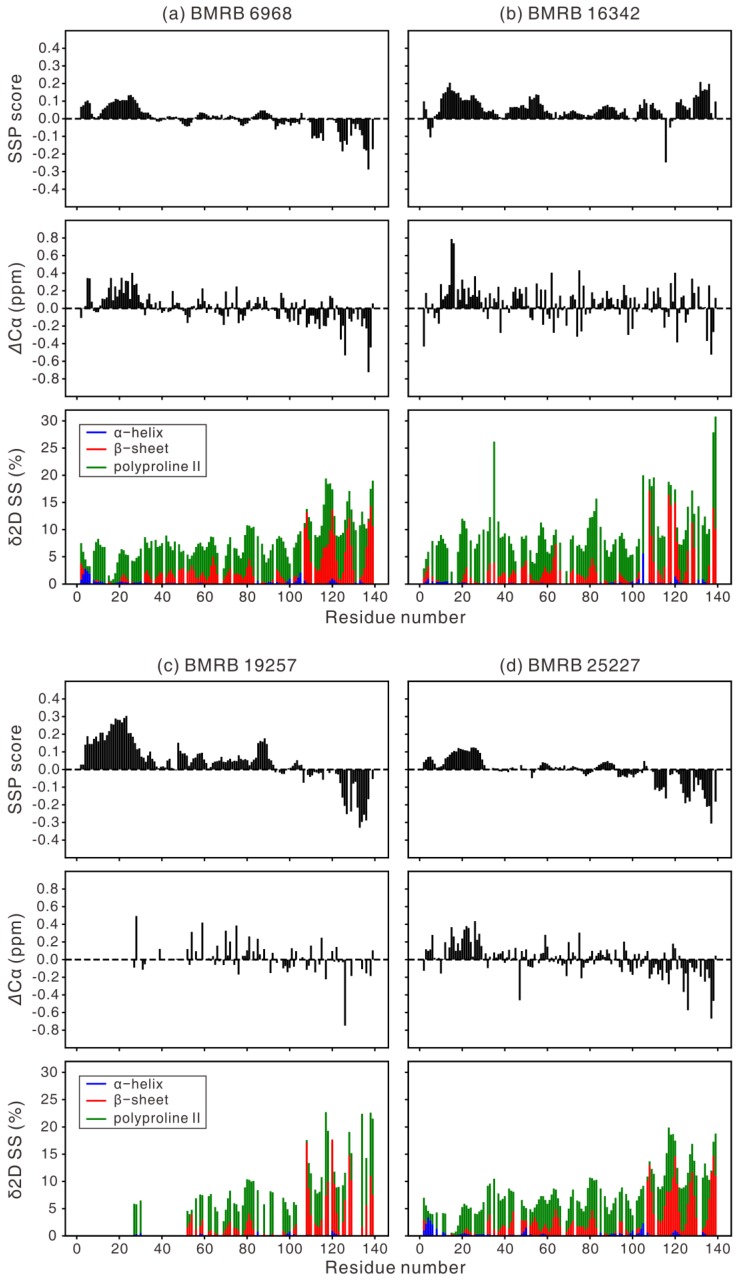
Comparison of the secondary structure propensity (SSP) scores, Cα secondary chemical shifts, and the secondary structure (SS) population from the δ2D analysis. The same ^13^C chemical shifts were used as an input for the analysis. As at least three chemical shifts are required for the δ2D analysis, the δ2D percentage is not displayed for a residue, if even one of the Cα, Cβ, and C′ chemical shifts was unavailable from the BMRB database (e.g., at least one ^13^C chemical shift information is missing for most residues of 1–50 in BMRB 19257, there was no C′ and Cβ chemical shift information for all residues in BMRB 26557 and 27348, respectively). Sample conditions for the six different NMR studies on α-synuclein (see Table 1) are (**a**) 1 mM αS, 20 mM phosphate, 0.5 mM EDTA, 200 mM NaCl, 10% D_2_O, pH 6.5, 285.5 K (BMRB 6968), (**b**) 0.3 mM αS, 20 mM NaOAc, 100 mM NaCl, 10% D_2_O, pH 3.0, 288 K (BMRB 16342), (**c**) 1.7 mM αS, 10% D_2_O, 90% H_2_O, 277 K, in-cell condition (BMRB 19257), (**d**) 0.5 mM αS, 10 mM sodium phosphate, pH 7.5, 100 mM NaCl, 5% D_2_O, 283 K (BMRB 25227), (**e**) 0.43 mM αS, 20 mM HEPES, 10% D_2_O, pH 7.0, 277 K (BMRB 26557), and (**f**) 1 mM αS, 200 mM NaCl, 0.5 mM EDTA, 20 mM phosphate, pH 6.5, 315 K (BMRB 27348).

**Figure 2 biomolecules-10-00428-f002:**
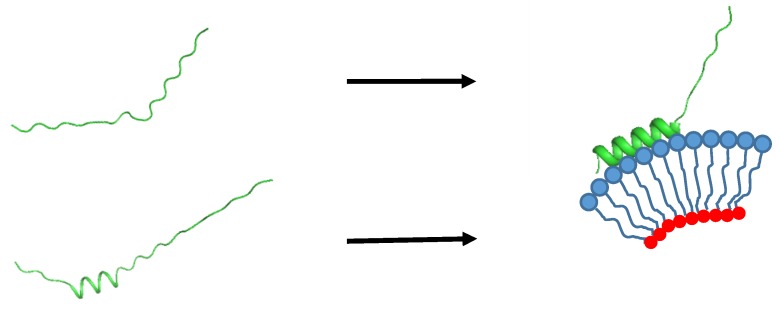
An illustration of the lipid-binding mode of αS. (top) Induced fit, (bottom) conformational selection. In the latter, an N-terminal helix is inserted merely to indicate the fact that a transient helix is present at the N-terminus of αS, i.e., the location of the helix is not exact. The thickness of the helix ribbon is adjusted to reflect the population of the helix.

**Table 1 biomolecules-10-00428-t001:** A list of nuclear magnetic resonance (NMR) studies on human αS. Seven reports deposited the assigned chemical shifts to the BMRB database, as shown in the third column.

Year	Sample Condition	BMRB	ref
2001	~100 μM αS, 100 mM NaCl, 10 mM Na_2_HPO_4_, pH 7.4, 283 K		[28]
2003	0.3 mM αS, 20 mM sodium phosphate, 50 mM SDS, pH 7.4, 298 K	5744 ^d^	[73]
2006	1 mM αS, 20 mM phosphate, 0.5 mM EDTA, 200 mM NaCl, 10% D_2_O, pH 6.5, 285.5 K	6968	[74]
2008	0.2 mM αS, PBS buffer, pH 7.4, 263 K		[35]
2009	0.65 mM αS, 10 mM phosphate, 140 mM NaCl, pH 2.5, 10% D_2_O, 288 K		[75]
2009	0.3 mM αS, 20 mM NaOAc, 100 mM NaCl, 10% D_2_O, pH 3.0 & pH 7.4, 288 K	16342	[76]
2009	0.6 mM αS, 20mM Na_2_HPO_4_ (pH 6.0), 6% D_2_O, 0,02% NaN_3_, in phospholipids, 293 K		[27]
2010	0.6 mM wild-type αS, mutants (A30P, E46K, A53T), 20 mM Na_2_HPO_4_, pH 6.0, 6% D_2_O, 0.02% NaN_3_, in phospholipids, 293 K		[77]
2012	0.1 mM αS, 5 mM dioxane, 20 mM sodium phosphate buffer, pH 6, in phospholipids, 288 K		[78]
2013 ^a^	1.7 mM αS, 10% D_2_O, 90% H_2_O, pH 6.2, 277 K	19257	[79]
2013 ^b^	- mM αS, 20 mM Tris-HCl, pH 7, 100 mM NaCl, 10% D_2_O, 288 K		[80]
2014	0.35 mM αS, 20 mM sodium phosphate, pH 6, 288 K		[81]
2014	50 μM αS, NaCl/sodium phosphate buffer, 5% glycerol, 288 K		[82]
2014	0.3 mM αS, 20 mM NaOAc, 100 mM NaCl, 10% D_2_O, pH 3.0 & pH 7, 288 K		[83]
2015	0.5 mM/0.7 mM wild-type/H50Q αS, 10 mM sodium phosphate, pH 7.5, 100 mM NaCl, 5% D_2_O, 0.01% NaN_3_, 0.001% DSS, 283 K	25227	[84]
2015	~0.43 mM (6.6mg/mL) αS, 20 mM HEPES, 10% D_2_O, pH 7.0, 277 K	26557	[85]
2016 ^a^	0.4 mM αS, 20 mM sodium phosphate, 150 mM NaCl, pH 7.0, 283 K		[60]
2017 ^c^	75 μM αS, PBS buffer, 0.02% NaN_3_, pH 7.4, 310 K		[86]
2018	~1 mM αS, pH 5.0, 10% D_2_O, 298 K		[87]
2018	1 mM αS, 20 mM phosphate, 200 mM NaCl, 0.5 mM EDTA, pH 6.5, 285.5 K, 295 K, 305 K, 315 K	27348	[88]

^a^ In-cell NMR. ^b^ No information is available on the αS concentration. ^c^ Aggregation inhibition experiment. ^d^ BMRB 5744 is for a folded αS, and hence is not used for the calculation of secondary structure propensity (SSP) values.

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
