# Peer review of "Salient Features of Monomeric Alpha-Synuclein Revealed by NMR Spectroscopy"

_biomolecules, 2020, doi:10.3390/biom10030428_

Round 1

Reviewer 1 Report

This review by Do-Hyoung Kim addresses the study of alphasyn by NMR. In an introduction paragraph, the interest to study alphasyn conformation is presented, in relation to alphasyn tendency to oligomerize in parkinson disease. In a second paragraph, transient secundary structures found in most of the IDPs are described. Examples are presented. The NMR parameters useful to detect these transient structures are listed. Some tools available to generate random coil chemical shifts or ensemble of conformations are cited. A comparison of studies of alphasyn, dating back 15 years, are referenced. Based on carbon chemical shifts provided by these studies, secondary structure propensity of alphasyn are compared and discussed. Following the observation of some inconsistencies, experimental parameters of data acquisition are discussed. The review is of interest to learn about IDPs and their study by NMR, and more specifically on alphasyn. Manuscript is well written.

I suggest to add NMR in the title as this review is strongly focused on the NMR technique.

« Conformation » is stated in the title, but only local secundary structure is described in detail in the manuscript. This is a restrictive view of IDP conformations. Ensemble conformation is also an important aspect, linked to oligomerization that might also depend on the exposure of the NAC region (modulated by PTMs or mutations). 3D conformation propensity and building of ensemble is also explored using NMR (PREs) and I suggest to add a paragraph dealing with exploration of Syn ensemble by use of NMR. I believe that would be of interest both to cover the aspect of alphasyn conformation but also as the manuscript is advertising the use of NMR, for the non-specialists to be able to grasp the full extend of its potential to study IDPs.

Some detail :

Page 2 of 12 : various theories : not clear – theories about what point ? Oligomerisation ? Pathogenesis ? Syn Function ? Np point fits well with the following enumeration :

ranging from pore formation followed by membrane leakage [17,18] to receptor-mediated mechanisms [19,20] and cellular  protection by binding with extracellular chaperones [21] are currently being presented.

Please rewrite.

KTKEGV motif that binds to synthetic lipid vesicles or detergent micelles in vitro and adopts a highly helical conformation [26-28] when bound ? Sentence could be more precise.

Reviewer 2 Report

I think that a review that brings all of the structural studies on alpha-synuclein together is a good idea and well described in the introduction. However by the time I reach the end of the paper I am left feeling that the authors are highlighting the issues with all of the studies (justified to make the reader aware to treat them with caution) however they base this conclusion on one single computational tool. The authors themselves state that there are many tools to investigate SSP however only use one, this manuscript would greatly benefit from a comparison using several and a description of what level of consensus there is between them. Particularly the effect of including additional parameters as can be done in POTENCI. Another key area of relevance to alpha-synuclein is interaction with lipids/membranes as it is likely that this represents the physiological environment so cannot be ignored, this should be covered in this review as there are many NMR studies carried out in the presence of lipids/mimetics that are highly relevant to understanding alpha-synuclein monomeric/native structure. The review would also benefit from a summary figure highlighting the conclusions found.

Additional points:

-line 65 should contain the word are not is

-line 75 is grammatical incorrect

-line 82 it is not clear what is meant by promote intramolecular interaction.

-define PPII at first mention

-line 125-126 this is the only mention of bicelles etc and feels out of place here, instead this needs expansion as described above

-line 196, the phrase in case we are dealing with mammalian cells. This manuscript details the cells used so this is known to the reader/author. Also the author mentions the effect/relevance of considering acetylation state is this known for the other NMR studies too? If so can this be commented on with relation the data shown in Figure 1. 
